# Molecular Detection of Tetracycline-Resistant Genes in Multi-Drug-Resistant *Escherichia coli* Isolated from Broiler Meat in Bangladesh

**DOI:** 10.3390/antibiotics12020418

**Published:** 2023-02-20

**Authors:** Gazi Sofiul Alam, Mohammad Mahmudul Hassan, Md. Ahaduzzaman, Chandan Nath, Pronesh Dutta, Hamida Khanom, Shahneaz Ali Khan, Md Ridoan Pasha, Ariful Islam, Ricardo Soares Magalhaes, Rowland Cobbold

**Affiliations:** 1Faculty of Food Science and Technology, Chattogram Veterinary and Animal Sciences University, Chattogram 4225, Bangladesh; 2Faculty of Veterinary Medicine, Chattogram Veterinary and Animal Sciences University, Chattogram 4225, Bangladesh; 3Queensland Alliance for One Health Sciences, School of Veterinary Science, The University of Queensland, Gatton, QLD 4343, Australia; 4EcoHealth Alliance, New York, NY 10018, USA; 5School of Veterinary Science, The University of Queensland, Gatton, QLD 4343, Australia

**Keywords:** poultry meat, AMR, *Escherichia coli*, tetracycline resistance, food safety, zoonoses

## Abstract

This study aimed to estimate the antimicrobial resistance (AMR) patterns and tetracycline-resistant gene profiles of *Escherichia coli* (*E. coli*) from broiler meat and livers sourced from live bird markets (LBMs) and supermarkets (SMs) in Chattogram, Bangladesh. In total, 405 samples were collected from SMs and LBMs, comprising muscle (n = 215) and liver (n = 190) samples. Disc diffusion tests were used to determine antimicrobial susceptibility profiles. PCR was used to identify *E. coli* and tetracycline-resistant genes. Over half (57%) of the chicken product samples were positive for *E. coli*. The AMR profiling of these isolates showed that the highest prevalence of resistance was against sulphamethoxazole–trimethoprim (89%), followed by tetracycline (87%), ampicillin (83%), and ciprofloxacin (61%). Among the antimicrobials listed by the World Health Organization as critically important, *E. coli* isolates were found to be resistant to cephalexin (37%), gentamicin (32%), and colistin sulfate (21%). A large proportion of *E. coli* demonstrated multi-drug resistance (MDR). Most (84%) of the tetracycline-resistant isolates encoded *tet*A. Of the remaining isolates, 0.5% encoded *tet*C, 6.0% encoded two genes, and 3.6% of isolates were *tet*D, which was newly identified by this study in Bangladesh. Broiler products in Bangladesh are frequently contaminated with multi-drug-resistant *E. coli*, with differential carriage of tetracycline genes. The prevalence of tetracycline resistance among *E. coli* indicates a concern for poultry health and welfare regarding the management of colibacillosis. It also indicates growing public health risks of AMR among broiler-associated pathogens, which can be transferred to humans via the food chain. Appropriate control measures should be developed and implemented, focused on the rational use of antimicrobials in poultry farming systems, to mitigate risk from this drug-resistant zoonotic pathogen from foods of animal origin and to protect public health.

## 1. Introduction

*Escherichia coli* (*E. coli*) is a Gram-negative bacterium that significantly impacts human and animal health [1]. Most *E. coli* reside in the large intestine of humans and animals as commensal bacteria, but some strains (pathotypes) represent dangerous pathogens that can result in intestinal and systemic illness under various conditions [2]. Antimicrobials are commonly used in veterinary medicine as part of a response to colibacillosis infection [3]. In many countries, antimicrobial agents are routinely fed to livestock, especially poultry broilers, in prophylactic form or as antimicrobial growth promoters (AGPs) [4]. The indiscriminate use of antimicrobials for production and preventive purposes is recognized to increase the risk of resistance among *E. coli* and other enteric pathogens [5]. The development of antimicrobial resistance (AMR) is a complex phenomenon involving various bacterial genetic and metabolic mechanisms and is expedited through antibiotic selection pressure [6,7]. In intensive broiler production in some countries, antibiotic selection pressure for resistance in microorganisms is high, and consequently, poultry fecal flora contains a comparatively high proportion of resistant microorganisms [8,9]. Those resistant organisms can be transmitted to humans via the food chain and through cycling to other animals and the environment [10].

It is known that poultry meat can be frequently contaminated with *E. coli* during the unhygienic handling and dressing of carcasses and meat [11]. Poultry meat is a potential source of human *E. coli* infection, as there is a higher chance of exposure via direct contact during food preparation or poor cooking [12]. Beyond being an important pathogen in their own right, the contamination of foods of animal origin with *E. coli* indicates poor hygiene practices and the potential presence of other zoonotic and pathogenic enteric bacteria such as *Campylobacter* and *Salmonella* [13]. Broiler meat has been previously demonstrated to carry antimicrobial-resistant *E. coli*, *Campylobacter* spp., and *Salmonella* spp., which are transmitted to humans via the food chain [14].

Research has shown that avian *E. coli* is prevalent in broiler meat worldwide. However, there is a lack of specific data for Bangladesh. A review article showed that most avian *E. coli*-associated AMR studies were conducted in selected metropolitan cities, namely Dhaka, Rajshahi, and Mymensingh [15]. Despite the presence of several veterinary and research laboratory facilities in these parts of the country, there is no systemic and structured surveillance for zoonotic AMR bacteria, despite the high risks for these pathogens, due to several factors. For example, in Bangladesh, the majority of farmers (>60%) use antibiotics without any prescription [16]. Besides farms, poultry meat vendors in live bird markets (LBMs) also use different types of antibiotics purely prophylactically to prevent unwanted bird mortality [17]. Tetracycline is the most frequently used antimicrobial due to its lower price and easy availability, which indicates a specific risk for the tetracycline-resistant strains of *E. coli* in Bangladesh [18]. Several other antimicrobials, such as ciprofloxacin, amoxicillin, and gentamicin, are also used at different stages of production. Therefore, this study was conducted to understand the current status of poultry product contamination with *E. coli* and associated resistance patterns in chicken meats sourced from the main two outlet types (local supermarkets and live bird markets) in the Chattogram Metropolitan Area (CMA), Bangladesh. A specific molecular investigation was carried out on the carriage of *tet*A, *tet*B, *tet*C, and *tet*D genes.

## 2. Materials and Methods

### 2.1. Study Area

The study was conducted in the Chattogram Metropolitan Area of Bangladesh from October 2020 to February 2021. Samples were collected from five supermarkets (SMs) and nine live bird markets (LBMs) (Figure 1). These two types of retail establishments were chosen as they represent the primary sources of poultry meat in Bangladesh, with most people buying poultry from LBMs, apart from in major urban areas. Although poultry products are commonly purchased from either retailer type, they are ultimately sourced from the same wholesalers. A key difference is that the further management of birds (including treatment with antimicrobials) and the processing of poultry at LBMs occur on-site, whereas SM processing is undertaken elsewhere.

### 2.2. Sample Size

A total of 405 poultry product samples were collected from 215 birds. We calculated the desired sample size using the online sample size calculation tool, Open Epi version 3.1, and based on the following equation [19]:Sample size (n) = [DEFF × Np(1 − p)]/[(d2/Z21 − α/2 × (N − 1) + p × (1 − p)]

We hypothesized the anticipated frequency of the outcome factor in the population (p) as 76.1% +/− 5% error, based on a previous study [20]. The design effect was set as 1 in this study. The Appendix A provide specific details for sample numbers by individual premises (Appendix A).

### 2.3. Sample Collection, Transportation, and Processing Procedures

Using appropriate hygiene procedures, samples were collected in separate, sterile, ziplocked bags. After collection, the samples were transported at refrigeration temperature to the Department of Microbiology and Veterinary Public Health (DMVPH), CVASU, for further investigation. The samples were diced and transferred to separate sterile test tubes containing buffered peptone water (BPW; HIMEDIA, Mumbai, India) and incubated at 37 °C overnight for primary enrichment.

### 2.4. Microbiological Isolation

#### 2.4.1. Isolation and Identification of *E. coli*

The enriched culture was streaked on a MacConkey agar medium (HIMEDIA) and incubated at 37 °C for 24 h. The colonies demonstrating appropriate *E. coli* morphology were streaked on eosin methylene blue (EMB) agar plates (Merck, Rahway, NJ, USA) and incubated at 37 °C for 24 h for the confirmation of colony morphology. Confirmed isolates were inoculated on blood agar and incubated at 37 °C for 24 h for DNA extraction. All phenotypically positive isolates were subjected to molecular identification with species-specific multiplex PCR with a DLAB Scientific (City of Industry, CA, USA) thermal cycler using primers for the *uid*A gene and the flanking region of the *usp*A gene [21].

#### 2.4.2. Screening of *E. coli* Isolates for Antimicrobial Resistance

*E. coli* isolates were screened for antimicrobial susceptibility using the Kirby–Bauer disc diffusion method [22]. Eight antimicrobials from six different classes (β-lactam antibiotics (including penicillins), tetracyclines, polymyxins, aminoglycosides, quinolones, and sulfonamides) were selected for screening based on human health importance and prevalence of use in the Bangladeshi poultry industry. The following antimicrobial agents (with respective disc potencies) were used: colistin (CT, 10 µg); tetracycline (TE, 30 µg); gentamicin (CN, 10 µg); doxycycline (DO, 30 µg); ampicillin (AMP, 10 µg); cephalexin (CL, 30 µg), sulfamethoxazole–trimethoprim (SXT, 25 µg); and ciprofloxacin (CIP, 5 µg). The interpretation of the results was based on the guidelines of CLSI (2020) (Appendix A). Due to a lack of CLSI disc diffusion standards for colistin sulfate, the results were interpreted based on the guidelines of OXOID [23].

#### 2.4.3. Polymerase Chain Reaction (PCR) for Tetracycline-Resistant Genes

All tetracycline-resistant *E. coli* isolates were further investigated using PCR. The isolates were recovered from storage via incubation on blood agar before the isolation of DNA using the boiling method. PCR reactions for *tet* genes were conducted using a DLAB Scientific thermal cycler. The primer sequences used for the PCR to detect *tet*A, *tet*B, *tet*C, and *tet*D genes were as described by Koo et al. (2011) [24]. PCR products were visualized using 1.5% agarose gel electrophoresis.

### 2.5. Preservation of the Isolates

All *E. coli* isolates were cultured in brain–heart infusion (BHI) broth and incubated overnight at 37 °C. For each isolate, 700 µL BHI broth culture was added to 300 µL of 15% glycerol in an Eppendorf tube. The tubes were stored at −80 °C for further investigation.

### 2.6. Statistical Analysis

All data were recorded and sorted (according to sample and market type) in Microsoft excel 2019 (Microsoft Corporation, Redmond, WA, USA) for statistical analysis. Descriptive statistics and statistical analyses were performed using STATA-13 (StataCorp, College Station, TX, USA). Univariate analysis was performed for the different antimicrobials tested for different markets. Mixed effect logistic regression was conducted using R software (R Foundation for Statistical Computing, Vienna, Austria). The isolates were defined as having a multi-drug-resistant (MDR) phenotype when they demonstrated resistance to three or more different antimicrobials.

## 3. Results

### 3.1. E. coli Contamination of Meat Samples

The putative *E. coli* isolates confirmed by culture also tested positive in the PCR. Among the 405 samples, 229 (56.5%; 95% CI 51.56–61.43%) demonstrated confirmed *E. coli* contamination, with 78% of the birds having an *E. coli*-positive sample. *E. coli* was found in 59.5% (n = 113) of liver samples and 54.0% (n = 116) of muscle samples (*p* = 0.2634). There was a higher, albeit not statistically significant, trend in the prevalence of *E. coli* in the meat samples collected from live bird markets (58.3% (n = 105)) than those from the supermarkets (55.1% (n = 124)) (*p* = 0.5157).

### 3.2. Antimicrobial Resistance Patterns of Poultry E. coli Isolates

According to the breakpoint guidelines of CLSI-2018, a significant percentage of *E. coli* isolates demonstrated resistance to the tested antimicrobials. Most significant among these was resistance to sulphamethoxazole–trimethoprim (88.7%), tetracycline (86.9%), ampicillin (82.5%), and ciprofloxacin (60.7%). The full resistance patterns of the isolates are shown in Table 1 and Figure 2. Each antimicrobial is listed per its designation as highly or critically important to human health by the World Health Organization [25]. The resistance profiles for all the tested antimicrobials were similar for the samples collected from both supermarkets and live bird markets (Table 2 and Figure 2), except for ciprofloxacin and doxycycline, which were significantly higher in the samples obtained from live bird markets (*p* = 0.0065).

### 3.3. Multi-Drug Resistance

Most of the isolated *E. coli* were resistant to at least three or more antimicrobials. The most common antimicrobials included within multi-drug resistance profiles were sulphamethoxazole–trimethoprim; tetracycline; ampicillin; ciprofloxacin; cephalexin; gentamicin; and colistin sulfate. The MDR phenotype was 4.27 times higher for LBM samples than for supermarket samples (Table 3; Figure 3). There was no statistically different rate for the MDR phenotype based on sample type.

### 3.4. Tetracycline-Resistant Genes

The PCR detection of tetracycline-resistant genes in phenotypically tetracycline-resistant isolates indicated no significant differences in the respective gene prevalence when sample type and source were considered (Table 4). Among the isolates, 84.4% (n = 168) encoded *tet*A, 5.0% (n = 10) encoded *tet*B, 3.0% (n = 6) encoded *tet*D, and 0.5% (n = 1) encoded *tet*C. Combinatorially, 3.0% (n = 6) encoded *tet*A + *tet*B, 0.5% (n = 1) encoded *tet*A + *tet*C, 2.5% (n = 5) encoded *tet*A + *tet*D, and 0% encoded *tet*B+ *tet*C, *tet*B + *tet*D, and *tet*C + *tet*D. No isolates encoded more than two resistance genes, while 12.1% (n = 24) of the isolates tested negative for all four resistance genes.

## 4. Discussion

AMR is a critical human and animal health issue worldwide, resulting in the progressive loss of antimicrobials’ efficacy in the face of an increasing abundance of exposure to resistant bacteria, with fewer new classes of antimicrobials being developed to fill gaps [25]. The current study’s findings revealed that the prevalence of *E. coli* from the breast muscle and livers of the broilers collected from supermarkets and live bird markets in Bangladesh is high, with the isolates frequently exhibiting resistance to multiple antimicrobial classes. The overall prevalence of *E. coli* in broiler chicken samples was similar to previous studies, where it was reported as 49–53% in Bangladesh [26], 66.3% in India [27], 66.8% in Sri Lanka [28], and 50.5% in Korea [29]. The prevalence of *E. coli* and resistance patterns of the isolated bacteria did not significantly differ based on the source of the samples. This finding is supported by the study of Ranasinghe et al. [28], which demonstrated similar results in Sri Lanka. This is likely because poultry supply networks share a common source, namely the wholesale markets for both LBMs and SMs, despite some differences in downstream processing. In this study, the overall prevalence of *E. coli* in broilers from LBMs was similar to those of Hossain et al. [30], who reported the prevalence was 63.6%, whereas Jakaria et al. [31] found 82% prevalence, and Bashar et al. [32] found 100% prevalence of *E. coli* in poultry. The presence of *E. coli* in the meat samples of SMs in our study was similar to that reported in a previous study, where it was recorded as 66.7–76.1% in chicken meat [20]. This study confirms the presence of relatively high levels of *E. coli* contamination, indicating the poor hygienic status of poultry meat in Bangladesh, and the public health risks of food-borne pathogens from these products. In this study, we found a higher prevalence of *E. coli* in muscle samples than in liver samples, which is supported by the findings of Ranasinghe et al. [28]. Generally, pathogenic bacteria are not present in the muscle tissues of healthy living birds [33]. However, with faults during slaughtering and meat processing, there can be contamination with the bacteria from the ingesta and surroundings [34,35]. In most of the LBMs, poultry meat handlers do not use gloves and do not practice proper hand washing during the processing of poultry meat, which is a major source of contamination for poultry meat [28].

The culture and sensitivity testing of the isolates showed the highest prevalence of *E. coli* resistance was for the combination of sulphamethoxazole–trimethoprim, followed by tetracycline and ampicillin. Most poultry-derived *E. coli* were sensitive to colistin, gentamicin, and cephalexin. Our findings are supported by the study of Parvin et al. [20], where the percentage of *E. coli* resistance against ampicillin, sulphamethoxazole–trimethoprim, and tetracycline was reported as 84.9–89.5%, and there was less resistance to gentamycin, colistin, and cephalexin (ranging from 8.1% to 46.5%). However, resistance to these antimicrobials was still substantive and represents a significant public health concern due to the importance of these antimicrobials in human medicine. Different studies have shown that poultry farms and their environments, such as manure and wastewater, frequently harbor AMR and MDR bacteria and antimicrobial residues, representing ongoing sources of selection pressure [36,37]. Vegetables and animal products that are used to make poultry feed collected from wet markets and shops [38,39] have also been identified as hot spots for AMR, providing evidence of direct linkage to human exposure [40]. Lack of knowledge of antimicrobial agents and indiscriminate practices of antimicrobial use in animal production without following the prescriptions of registered veterinarians in poultry farms are the likely sources of AMR bacteria in the meat of LBM and SM samples [41,42,43]. As accurate antimicrobial dosages cannot be maintained according to age and body weight, animals’ commensal flora is exposed to low residual doses over prolonged periods, which enhances the development of resistance [44]. The high prevalence of antimicrobial residues in the tissues and environments of the farms enhances the public health risk of AMR in developing countries such as Bangladesh [45,46]. Horizontally transferred resistant bacteria and associated genes have emerged in those farms from antimicrobial residues [47,48].

In Bangladesh’s poultry sector, colibacillosis is a commonly encountered disease and represents one of the most usual indications for antimicrobial use [49,50]. Hence, antimicrobials are often used in the whole flock at the time of infection or prophylactically to prevent infections [51] and are used regularly as a growth promoter at a lower dose to increase profitability [52]. This situation indicates multiple pathways for developing resistance genes, which will be transferred to other pathogens horizontally and vertically [53].

MDR *E. coli* were commonly observed in this study. This aligns with the findings of Li et al. [54], who stated that 70.9% (N = 219) of their isolates were MDR (resistant to at least three classes of antimicrobials), as well as with the results from Ranasinghe et al. [28], who reported 82.6%, and Parvin et al. [20], who reported 100% rates of MDR in chicken meat. In comparison, only 6.5% (N = 20) of the isolates showed no observable resistance to the different groups of antimicrobials tested. Islam et al. [55] reported that 100% of AMR *E. coli* found in poultry were also characterized as being MDR. Results such as these are alarming since MDR pathogens are becoming significant food-borne pathogens that severely compromise treatment attempts for both humans and animals.

Tetracycline resistance was a specific focus of the current study due to the prevalence of the use of this drug in the poultry sector in Bangladesh [50], as well as the historically high levels of resistance being previously reported [20,56]. Previous studies in Bangladesh have mapped the changing patterns of tetracycline-resistant genes over time. Obeng et al. [57] found that *tet*A had the highest frequency, similar to the current study. Adelowo et al. [58] found that *tet*A was present in 21% of *E. coli,* which was lower than the present study, whereas *tet*B was present in 17% of isolates, which is comparatively higher than that found in this study. The current study confirms trends of the increasing prevalence of non-*tet*A genes amongst poultry *E. coli* isolates (specifically *tet*B and *tet*C) and is the first study to report the identification of *tet*D genes.

AMR is one of the most significant threats to veterinary and public health. All the antimicrobials screened for in the current study are recognized as highly or critically important to human health [59]. To mitigate the AMR health threat before it manifests as large-scale medical emergencies, scientific knowledge and science-based evidence are needed to identify risks and appropriate mitigation strategies [60,61]. At the same time, livestock profitability and sustainability can be increased through more effective agriculture practices that work along with reduced AMR development through the rational use of antimicrobials [62,63,64]. A recognized limitation of the current study is with respect to sample sizes. Although the total number of samples collected was in excess of the estimated necessary sample size (280), the number of individual birds sampled was less than this. Future studies should aim to be resourced such that they can accommodate larger sample sizes, as well as a broader array of antimicrobials. For instance, it would be valuable to screen for extended-spectrum β-lactamase genes. Resource limitations meant that the current study needed to focus on cephalexin, based on the commonality of its use in Bangladesh.

## 5. Conclusions

Interest in AMR associated with poultry and other food animals is sparked by a concern for human health being threatened by zoonotic pathogens and by selection for AMR determinants. The levels of *E. coli* contamination demonstrated in the current study indicate significant hygiene problems with production and processing in Bangladesh that exposes humans to chicken-associated fecal pathogens. Furthermore, the high rates of *E. coli* AMR against antimicrobial agents indicate significant reservoirs of resistance genes within the natural flora of poultry, and the biome of poultry-associated foods, representing an ongoing risk for resistance development among other enteric pathogens. Due to increasing human mobility and the globalization of food distribution chains, the impacts of resistance are not limited to individual countries, such as Bangladesh, in this case. Awareness of the threat of AMR, better controlling the use of antimicrobial drugs in animal production settings, and training programs emphasizing effective antimicrobial stewardship at a practical level are highly recommended to protect the food chain. Furthermore, enhanced food hygiene practices will reduce the risk of the spread of AMR enteric pathogens on food and to humans.

## Figures and Tables

**Figure 1 antibiotics-12-00418-f001:**
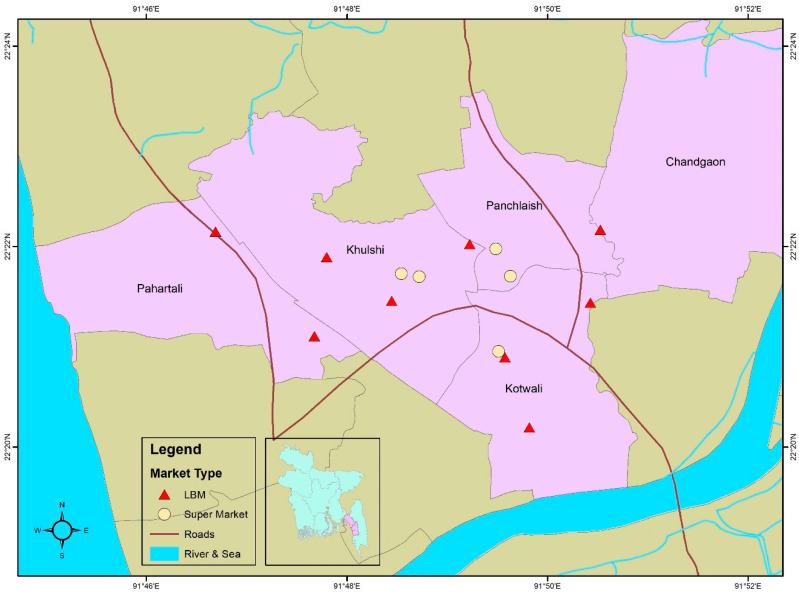
Map of the study area locating the LBM (live bird market) and SM (supermarket) sites sampled.

**Figure 2 antibiotics-12-00418-f002:**
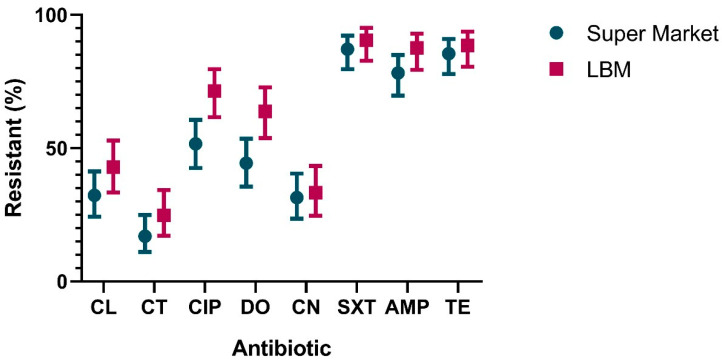
Antimicrobial resistance patterns for broiler-derived *E. coli* by market type (supermarket; LBM = live bird market). CL = cephalexin; CT = colistin; CIP = ciprofloxacin; DO = doxycycline; CN = gentamicin; SXT = sulfamethoxazole–trimethoprim; AMP = ampicillin; TE = tetracycline.

**Figure 3 antibiotics-12-00418-f003:**
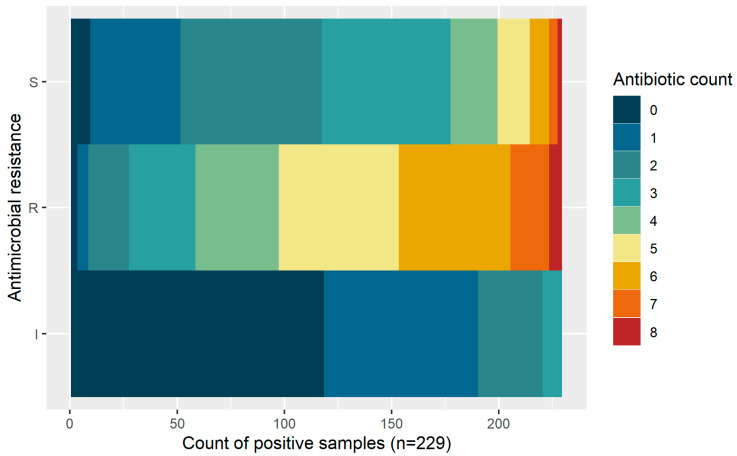
Patterns of multi-drug resistance for broiler-derived *E. coli* based on resistance classifications as susceptible (S), resistant (R), or intermediate (I).

**Table 1 antibiotics-12-00418-t001:** Patterns of antimicrobial resistance of poultry-derived *E. coli* isolates, relative to the WHO listing as highly and critically important.

Antimicrobial Agent and Disc Concentration	Number (%) of Intermediate (I) Isolates	Number (%) of Resistant (R) Isolates	Number (%) of Non-Susceptible (NS) Isolates (I + R)
Cephalexin (CL, 30 µg) ^a^	0 (0%)	85 (37.1%)	85 (37.1%)
Ampicillin (AMP, 10 µg) ^b^	12 (5.2%)	189 (82.5%)	201 (87.8%)
Tetracycline (TE, 30 µg) ^a^	14 (6.1%)	199 (86.9%)	213 (93.0%)
Doxycycline (DO, 30 µg) ^a^	65 (28.4%)	122 (53.3%)	187 (81.7%)
Gentamicin (CN, 10 µg) ^b^	22 (9.6%)	74 (32.3%)	96 (41.9%)
Ciprofloxacin (CIP, 5 µg) ^b^	42 (18.3%)	139 (60.7%)	181 (79.0%)
Colistin sulfate (CT, 10 µg) ^b^	0 (0%)	47 (20.5%)	47 (20.5%)
Sulphamethoxazole–trimethoprim (SXT, 25 µg) ^a^	4 (1.7%)	203 (88.6%)	207 (90.4%)

^a^ WHO-listed highly important antimicrobials. ^b^ WHO-listed critically important antimicrobials.

**Table 2 antibiotics-12-00418-t002:** Antimicrobial resistance patterns for *E. coli* isolated from different poultry products from different vendor types.

Source	Organ	Level of Resistance *	Antimicrobial Resistance Pattern, n (%).							
			CL	CT	CIP	DO	CN	SXT	AMP	TE
Supermarket	Liver (n = 56)	I	0 (0.0)	0 (0.0)	9 (16.1)	15 (26.8)	4 (7.1)	0 (0.0)	2 (3.6)	2 (3.6)
		R	15 (26.8)	7 (12.5)	28 (50.0)	29 (51.8)	13 (23.2)	52 (92.9)	45 (80.4)	48 (85.7)
		NS	15 (26.8)	7 (12.5)	37 (66.1%)	44 (78.6	17 (30.4)	52 (92.9)	47 (83.9)	50 (89.3)
	Muscle (n = 68)	I	0 (0.0)	0 (0.0)	16 (23.5)	23 (33.8)	4 (5.9)	1 (1.5)	5 (7.4)	3 (4.4)
		R	25 (36.8)	14 (20.6)	36 (52.5)	26 (38.2)	26 (38.2)	56 (82.4)	52 (76.5)	58 (85.3)
		NS	25 (36.8)	14 (20.6)	52 (76.5)	49 (72.1)	30 (44.1)	57 (83.8)	57 (83.8)	61 (89.7)
Live bird market	Liver (n = 57)	I	0 (0.0)	0 (0.0)	10 (17.5)	14 (24.6)	9 (15.8)	2 (3.5)	4 (7.0)	3 (5.3)
		R	29 (50.9)	13 (22.8)	41 (71.9)	37 (64.9)	15 (26.3)	52 (91.2)	49 (86.0)	53 (93.0)
		NS	29 (50.9)	13 (22.8)	51 (89.5)	51 (89.5)	24 (42.1)	54 (94.7)	53 (93.0)	56 (98.2)
	Muscle (n = 48)	I	0 (0.0)	0 (0.0)	7 (14.6)	13 (27.1)	5 (10.4)	1 (2.1)	1 (2.1)	6 (12.5)
		R	16 (33.3)	13 (27.1)	34 (70.8)	30 (62.5)	20 (41.7)	43 (89.6)	43 (89.6)	40 (83.3)
		NS	16 (33.3)	13 (27.1)	41 (85.4)	43 (89.6)	25 (52.1)	44 (91.7)	44 (91.7)	46 (95.8)

* I = intermediate; R = resistant; NS = non-susceptible; CL = cephalexin; CT = colistin; CIP = ciprofloxacin; DO = doxycycline; CN = gentamicin; SXT = sulfamethoxazole–trimethoprim; AMP = ampicillin; TE = tetracycline.

**Table 3 antibiotics-12-00418-t003:** Logistic regression for possession of MDR phenotype among *E. coli* based on broiler sample type.

		95% CI		
	OR	Lower	Upper	*p*-Value
Market Type				
Super Market	ref.			0.005
LBM	4.27	1.67	13.17	
Tissue Type				
Liver	ref.			0.785
Muscle	0.89	0.38	2.03	
Intercept	4.94	2.66	10.00	<0.001

OR: odds ratio; CI: confidence interval.

**Table 4 antibiotics-12-00418-t004:** Prevalence of tetracycline-resistant genes of *E. coli* in broiler meat samples.

Source	Organ	Tetracycline-Resistant Isolates	Prevalence of Tetracycline-Resistant Gene, n (%) (95% CI)			
			*tet*-A	*tet*-B	*tet*-C	*tet*-D
SM	Liver	48	43 (89.6) (77.34–96.53)	4 (8.3) (02.31–19.99)	0 (0)	2 (4.2) (0.5–14.25)
	Muscle	58	49 (84.5) (72.58–92.65)	4 (6.9) (1.91–16.73)	1 (1.7) (0.04–9.24)	1 (1.7) (0.04–9.24)
	*p*-value		0.4364	0.7804	0.3607	0.4504
LBM	Liver	53	41 (77.4) (63.8–87.72)	2 (3.8) (0.5–13.00)	0 (0.0)	3 (5.7) (1.18–15.66)
	Muscle	40	35 (87.5) (73.2–95.8)	0 (0.0)	0 (0.0)	0 (0.0)
	*p*-value		0.2103	0.2142	1.0000	0.1261
SM*LBM	*p*-value		0.3249	0.0821	0.3477	0.8706

SM = supermarket; LBM = live bird market.

## Data Availability

Data are available in Appendix A.

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
