# Peer review of "Molecular Detection of Tetracycline-Resistant Genes in Multi-Drug-Resistant Escherichia coli Isolated from Broiler Meat in Bangladesh"

_antibiotics, 2023, doi:10.3390/antibiotics12020418_

Round 1

Reviewer 1 Report

This study is well designed, presented and discussed.

L114-123 and L221 correct E coli

Fig.2 show all abbreviations used on the figures

The conclusion section should be shorten to show only the results of this work

Please correct the formation of microbial names for exaplele L352-355-365-375-378-391-401-409-415-420-422-429-431-etc

Please correct the formatting of journal names such as 451-462-464….etc

Reviewer 2 Report

The manuscript covers an interesting topic included in a One Health approach and food safety. The manuscript is well-written and structured.  Introduction section provides a good background on the topic and justify the research publication. M&M and Results section also are well-prepared. However, discussion is very generalist and did not proper discussed the main results obtained in the research. A deep reformulation on discussion section is required because the current version did not provide new insights on this topic and is widely unspecific. The same can be seen in conclusion section. Authors must provide a precious conclusion based on their results. Due the generalist characteristic from these both sections, I suggest major revision and require a new evaluation before acceptance.

Minor remarks

-        I suggest removing “tetA, tetB, tetC, 2 and tetD Genes” from title

-        How was sample size estimated? Please clarify this in M&M section

-        Line 221 – E. coli in italic

Reviewer 3 Report

The manuscript entitled “Molecular Detection of Tetracycline Resistance tetA, tetB, tetC, and tetD Genes in Multidrug-Resistant Escherichia coli Isolated from Broiler Meat in Bangladesh” may not be acceptable for publication. Some of the concerns are as follows.

1. The colistin sulfate disc was used according to the CLSI 2018 guidelines. The CLSI 2018 and even CLSI 2022 don’t have the criteria to interpret the colistin susceptibility using disc diffusion method. The MIC using broth dilution method is recommended method for colistin susceptibility.

2. Among the cephalosporins, only Cephalexin was tested. Why not third generation cephalosporins.

3. Why only tet genes why not others especially ESBLs? Any rationale for that?

4. Several sentences need restructuring such as “Data from this study indicated very high rates of common antimicrobials amongst the poultry samples collected”

”Culture and sensitivity testing of isolates showed the highest prevalence of E. coli resistance was against the combination of sulfamethoxazole-trimethoprim”

The right spellings for gentamycin are “gentamicin”

Round 2

Reviewer 2 Report

The discussion section has been improved and minor remarks were attended. However, the conclusion section remains improper. The conclusion is a repetition of results and discussion without proper conclude their research. This section MUST be improved as previously recommended. 

Reviewer 3 Report

The authors have explained "Due to a lack of CLSI disc diffusion standards for colistin sulfate, results were interpreted based on the guidelines of OXOID [23]. and have cited this article. 

Matuschek, E.; Å hman, J.; Webster, C.; Kahlmeter, G. Evaluation of five commercial MIC methods for colistin antimicrobial susceptibility testing for Gram-negative bacteria. Poster P0161. 27th Eur Congr Clin Microbiol Infect Dis, Vienna, Austria 2017, 22. 

The main concern is that the MIC breakpoints were not available in CLSI when this paper was published but now the CLSI 2022 have MIC breakpoints. 

The authors are performing the Disk Diffusion assay and are citing the paper describing the MIC breakpoints. 

Is there any Zone of Inhibition described in this paper ?

Have the authors performed MIC ? 

At least please add the Criteria (Zone of Inhibition) to classify any isolate as Susceptible or Resistant with suitable reference. 

Round 3

Reviewer 3 Report

The manuscript can be accepted for publication